# Encapsulation of Orange Oil Using Fluidized Bed Granulation

**DOI:** 10.3390/molecules27061854

**Published:** 2022-03-12

**Authors:** Gary Reineccius, Shardul Patil, Vaidhyanathan Anantharamkrishnan

**Affiliations:** Department of Food Science and Nutrition, University of Minnesota, St. Paul, MN 55018, USA; shardul933@gmail.com (S.P.); anant040@umn.edu (V.A.)

**Keywords:** encapsulation, spray drying, flavor, agglomeration

## Abstract

The primary objective of this research is to determine how granulation compares to spray drying/agglomeration for producing larger, more dense flavoring particles. Granulation can yield large, dense particles and thereby negate the need for a two-step process (spray drying followed by agglomeration) to achieve improved flow/handling properties of dry flavorings. In this study, a 55% solids slurry (blend of OSAn-modified starch and maltodextrin 15DE) was prepared and then single-fold orange peel oil was added at 20 or 25% of the carrier solids level. The 20% flavoring emulsion was spray dried (SD), and a portion of the resultant powder then agglomerated (Agg) in a bottom spray, fluidized bed. A second emulsion of the same carrier composition but using 25% orange oil based on carrier solids was prepared and subjected to fluidized bed granulation (FBG). Particle size, density, orange oil retention and oxidative stability on storage were determined. Overall, it is observed during this study that FBG produces orange oil encapsulates that possess better properties, such as more resistance to oxidation, a better retention of orange oil and a higher density than SD or SD/Agg powders.

## 1. Introduction

Encapsulation is a common approach used to maintain and protect flavoring materials. Encapsulation provides protection against the evaporation of flavor components and generally minimizes unwanted chemical reactions, primarily oxidation during storage [1]. A variety of methods are used in industry for flavor encapsulation, such as spray drying, spray chilling, coacervation or extrusion [2]. However, spray drying is the most common encapsulation process used in the flavor industry due to availability and the low cost of the spray drying process, a high retention of volatile components in the product and good shelf-life properties. However, spray-dried flavorings have some undesirable properties resulting from the small final particle size and low particle density. These properties result in poor flow in manufacturing, slow dispersion in liquid solutions (when used as an ingredient in manufacturing or by the consumer), and dusty conditions in packaging and handling. One approach to managing these undesirable properties involves agglomeration. The agglomeration of SD flavorings is most commonly performed by a rewet agglomeration process, i.e., the dry powder is fluidized and then rewet so the particles adhere to each other to form stable bridging between particles on drying. An increase in the particle diameter of up to 150 μm via agglomeration has been reported in the literature [3]. An alternative approach to produce larger, more dense articles is to use a single-step fluidized bed granulation (FBG) process.

FBG is a process which involves the conversion of atomizable liquids such as suspensions and emulsions into free-flowing granular solids by the inclusion of different processes such as wetting, drying, particle size enlargement and homogenization into a single step of processing [4]. The equipment used for FBG may be the same as that used for the agglomeration of fine powders (Figure 1). In this process, a fluidized bed of powder (an unflavored “seed” material) is formed by passing warm air upward through the seed material. This causes an intense movement of the seed particles inside the agglomerator and, hence, a fluidized bed is formed. An aqueous flavor emulsion (flavor system plus typical carrier solids) is then sprayed into the bed to rewet the fluidized seed bed. The atomized infeed emulsion collides with the fluidized seed bed to result in particle growth. The water from the fluidized infeed emulsion dries and leaves a slightly larger dry particle. This process of growth continues until the desired particle size is obtained. Then, the spraying is stopped, and the system run until the particles reach the desired dryness. A dust-free, easily flowing agglomerate is formed [5,6]. The residence time of the emulsion in a fluidized bed is controllable and, hence, the process can be continued until the desired particle size is obtained [6].

Burger et al. [6] investigated the application of FBG for producing flavor encapsulates. As previously described, FBG is carried out by the placement of inert seed material inside a fluidized bed followed by cycling between spraying flavor emulsion into the fluidized powder bed and then drying. Inlet air temperatures in the range of 60 to 100 °C were used and resulted in flavor being encapsulated in a glassy state, carbohydrate matrix. This process can be continued until the desired particle size is obtained. While there are numerous publications on the granulation of flavorings, Burger et al.’s [6] study is one of the few to report on the oxidative stability of granulated flavorings (e.g., orange oil). They found that using granulation with the proper blend of carrier solids (50% Capsul, 25% 20 DE maltodextrin and 25% sucrose) yields a powder with an excellent shelf life and physical properties.

Schleifenbaum et al. [7] was awarded a patent for a one-step fluidized bed encapsulation procedure to produce granulated flavorings. The target specifications were to have a mean particle size ranging from 0.2 to 2 mm, be dust free, have a flavor load from 1 to 25% and flavor retention in processing ranging from 60 to 90%. The target processing time should be less than 20 min, preferably from 2 to 15 min. These targets were accomplished by spraying a flavored emulsion into a fluidized bed containing seed particles with similar components as the spray-dried flavor base (maltodextrin and modified starch as carrier solids plus water plus flavoring). Different weight ratios of individual carrier solids were used to create emulsions of different flavors (e.g., strawberry and chicken). Nitrogen gas at 140 °C was used for fluidization and the granules created by this process could be coated further in the same instrument by other flavor emulsions or fats using similar encapsulation conditions. The encapsulates were found to have applicability and usage in products such as biscuits, ice cream, fruit jellies and tablets.

Benelli et al. [8] described an FBG method for producing a granulated rosemary extract flavoring. The encapsulates prepared by this method were compared with encapsulates manufactured by spray drying for parameters such as shelf life, polyphenol retention and bulk density. Cassava flour and sugar pellets were used as the seed material. It was observed that fluidized bed granules possessed a better flowability and higher retention of polyphenols than spray-dried encapsulates. The higher retention of polyphenols was postulated to be due to different drying mechanisms for the processes. In the SD process, the product dries due to direct contact with hot drying air, which results in a higher temperature of droplet than one would expect from FBG. Product residence time is higher in FBG than SD, but since the core material is renewed continuously, this appears to result in a lower loss of volatiles. Benelli et al. [8] found fluidized bed granules had a larger particle size, higher bulk density and better flowability than SD powder. Additionally, SD powders, unlike the granulated product, were observed to be sticky and had a tendency to agglomerate. The use of sugar pellets as a seed material produced particles with a smoother surface and spherical shape than cassava flour. A polyphenol retention of up to 100% was obtained using FBG.

The objectives of this current research are to compare SD, Agg and FBG processes for the production of a dry orange oil flavoring. Primary criteria for comparison are physical and chemical properties of the produced powders (particle size, density, flavor retention and oxidative stability).

## 2. Materials

ACS-grade acetone and 4-heptanone were obtained from (Sigma Aldrich Inc., St. Louis, MO, USA). Anhydrous methanol (Avantor, Center Valley, PA, USA), anhydrous sodium sulfate (Fisher Scientific, Fair Lawn, NJ, USA) and the chemicals comprising the pyridine-free reagents kit (Photovolt Instruments Inc., Minneapolis, MN, USA) for the Karl Fischer titration system were used for Karl Fischer moisture analysis. Magnesium chloride hexahydrate salt was used to create saturated salt solutions for the adjustment of sample water activity (Sigma Aldrich, St. Louis, MO, USA). Single-fold orange peel oil (Firmenich, Princeton, NJ, USA) was used as the primary encapsulation load material. Maltrin M-150^TM^ (Grain Processing Corp., Muscatine, IA, USA) and Capsul^TM^ (octenyl succinic anhydride, OSAn, substituted modified starch) (Ingredion, Westchester, IL, USA) were used as carrier materials.

### 2.1. Preparation of Orange Oil Emulsion

The carrier materials (Maltrin M-150 and Capsul) were dissolved in water at ambient temperature and allowed to stand overnight. Immediately before spray drying, orange oil was added to carrier solid and water mixture, and mixed using a high shear blade mixer (Greerco Corp., Hudson, NH, USA) for 15 min. The compositions of the emulsions are shown in Table 1.

### 2.2. Spray Drying

An APV spray dryer with centrifugal wheel atomization was used in this study. This emulsion (7285 g) was spray dried using an inlet temperature of 200 °C and exit temperature of 100 °C. The evaporation capacity of the dryer was approximately 15 kg/h under these air temperatures. This dryer used a centrifugal wheel atomizer operated at 25,000 RPM.

### 2.3. Fluidized Bed—Agglomeration

The spray-dried orange oil was agglomerated using a Glatt GmbH Systemtechnik D-01277—GPCG 1- Bottom spray Wurster (Glatt Air Technologies, 20 Spear Rd., Ramsey, NJ 07446) 2 separate batches differing in run times (30 and 90 min). The inlet air temperature was set at 80 °C and it was cycled at an exit air temperature range 47–54 °C (product temperature range 44–50 °C).

### 2.4. Fluidized Bed—Granulation

The 25% orange oil emulsion was sprayed into the fluidized bed described above. Two hundred grams of maltodextrin was used as seed material. Three different batches with different run times were carried out—25, 35 and 45 min.

### 2.5. Storage of Encapsulated Powders

Immediately after production, approximately 15 g of each powder was placed in an RH-controlled desiccator (contained saturated Magnesium chloride hexahydrate solution (0.33 aw), evacuated and filled 4 times with dry N_2_ to create an oxygen-free atmosphere. The desiccator was kept in the dark at ambient temperature until the powder tested to aw = 0.33 +/− 0.02 (up to 4 weeks). When the desired aw was reached, the desiccator was opened to allow oxygen to enter and then was moved into a walk-in incubator (35 °C), and each powder was sampled weekly for degree of oxidation (limonene oxide content). The method was adopted from Anker and Reineccius [9].

### 2.6. Particle Size of Encapsulated Powders

A Horiba LA-960 Laser Particle Size Analyzer (HORIBA Scientific, Edison, NJ, USA) was used to obtain the particle size distribution of each of the prepared powders. Approximately 0.5 g of each powder was added to the instrument. Instrument settings were as follows—air pressure: 0.15 mPa; feeder: 100% (automatic); refractive index: 1.67. Particle size was analyzed in duplicate. Mean particle size (volume mean diameter of powder particles) of the powders was recorded.

### 2.7. Moisture Content by the Karl Fischer Method

The moisture content of the water activity-adjusted powder samples was determined by the Karl Fischer method using a Metrohm KF 756 Aquatest CMA instrument (Photovolt Instruments Inc., Minneapolis, MN, USA). Approximately 0.20–0.25 g of sample was weighed into 20 mL headspace vials. Twenty grams of anhydrous methanol was weighed into the vial that contained the encapsulated material and the vial was capped and allowed to equilibrate overnight in a shaker. Approximately 1 mL of sample mixture was injected into the Aquatest CMA instrument and the Aquatest output reading was used to calculate the moisture content on a dry basis according to the manual instructions. Methanol blanks were prepared by weighing 20 g of methanol into a sample vial and determining the moisture in the solvent. All samples were analyzed in duplicate, including blanks.

### 2.8. Particle Density by Nitrogen Pycnometry

Particle density of powders was determined with a multi-pycnometer (QuantaChrome Corp., Boynton Beach, FL, USA). The pycnometer was calibrated according to the manual instructions. The large sample cylinder (cell) was used for determination of powder density. To determine the specific volume, 15 to 20 g of encapsulated material was accurately weighed into the sample cell of the instrument. Density analyses were performed in duplicate. True powder volume was calculated by using the following Equation (1):Vp = Vc − Vr {(P1/P2) − 1}(1)

Here: Vp—powder volume; Vc—large sample cell volume; Vr—large reference volume.

Density was calculated by the following Formula (2):Density = Mass of product inside sample cell/Vp.(2)

### 2.9. Orange Oil Retention

#### 2.9.1. Gas Chromatography

The quantity of orange oil in the prepared powders was determined by gas chromatography. Encapsulates (1.5 g) were dissolved in 8.5 mL of water and mixed well using a vortex mixer. An aliquot of this mixture (1.0 mL) and 4 mL of acetone solution containing 2-heptanome as ISTD were added with constant stirring. After mixing, the mixture was allowed to settle for one hour and a portion of the supernatant was transferred to 2 mL autosampler vials and loaded into a HP7673A automatic sampler (Hewlett-Packard, Wilmington, DE, USA). Two μL of each extract was automatically injected into an HP 5890 series II GC equipped with an HP-5MS capillary column (30 m × 0.25 mm × 0.25 μm) (J&W Scientific, Folsom, CA, USA) and a flame ionization detector (FID). A 1:50 sample inlet split ratio was used.

The amount of limonene present in the encapsulates was calculated based on the ISTD/limonene peak areas correcting for relative response ratios. The amount of orange oil retained was calculated by using the following Equation (3) assuming that limonene comprised 95% of the orange oil.
Volatile oil retention (%) = ((oil content obtained using GC peak areas)/(theoretical oil content)) × 100%(3)

#### 2.9.2. Limonene Oxidation by Gas Chromatography (GC)

Limonene oxide was also determined from the GC data. The oxidative stability of encapsulated orange oil was expressed as the ratio of peak areas of the sum of limonene oxide isomers to limonene (defined as mg LO/g L ratio in this research). The data for this calculation were obtained at the same time as the orange oil retention data described in the previous section. All the samples were analyzed in duplicate.

## 3. Results and Discussion

### 3.1. Particle Size

Under the operating conditions that were utilized in this research, FBG samples had the largest particle size values of all prepared encapsulates (Figure 2). While larger mean particles sizes than we produced via agglomeration could be achieved by extending the run time, agglomerates are typically weaker in structure than those produced by FBG and, thus, FBG can produce the largest particle sizes. FBG yields very dense/strong structures that are only limited in size by the ability to fluidize the bed particles.

Spray-dried samples offered the smallest mean particle size values (ca. 60 µm). A microscopic examination of the spray dried particles typically reveals some agglomeration for spray-dried powders so the mean particle size would be somewhat smaller without this inadvertent agglomeration. The agglomeration of the spray-dried orange oil increased the mean particle size from ca. 60 µm to 169 µm. (Note: there was no particular target in this study for particle size in agglomeration—the chosen particle size generally provides the desired physical properties with limited processing time.) These values, again, were a function of equipment run time, but were fairly consistent with other studies [10,11].

Benelli et al. [8] produced rosemary extract encapsulates with a large particle size by using fluidized bed granulation. Mean particle sizes obtained in their study ranged from 611 to 645 µm.

Particle size distribution data for three products (SD, Agg 90 min run and FBG 45 min run) are shown below in Figure 3. It can be observed from the graphs that the FBG sample had a more uniform particle size distribution when compared to the Agg and SD samples, as the graph had a single, clearly defined peak in contrast to the less finely defined peak of the SD sample and two visible peaks obtained for the Agg sample.

### 3.2. Particle Density Comparison

It can be observed from Figure 4 below that FBG produced samples with the highest particle density, followed by Agg and SD. Density values increased with an increased run time (Agg and FBG) of the process.

It was also found that Agg increased the density of the SD orange oil up to 22%. Similar density values in the range of 0.98 to 1.21 g/cc were reported by Buffo et al., although their carrier solid compositions were different [10].

### 3.3. C. Effect of Processing Method on the Retention of Orange Oil during Encapsulation

In the case of SD and Agg, flavor losses occurred both during the initial spray drying and secondary agglomeration steps. The granulated samples started with an unflavored seed material and the only flavor expected in the final product came from the use of a flavored emulsion to build the granules. Thus, the Gran samples started with 0% flavor and increased as the emulsion was added to the fluidized bed. Therefore, one would expect the Gran samples with the shortest processing time, i.e., 25 min, to have the lowest flavor load after granulation and the samples with the longest processing time to have the highest flavor load (as illustrated in Table 2).

The data shown in Figure 5 show the % of flavor retained in the final powders produced by spray drying, agglomeration and granulation. It can be seen from Figure 5 that the 25 min granulation sample had the highest percent of orange oil retention and the lowest flavor load of all samples. Several factors may have combined to produce this result. In terms of the granulated samples, this sample had the shortest residence time in the fluidized bed granulator and the lowest flavor level: both factors supported the low levels of flavor loss. One can see that more time spent in the fluidized bed (35 and 45 min) resulted in a greater flavor loss in the finished powders. Relative to spray drying and agglomeration, the operating temperatures used in granulation were substantially lower, thereby reducing flavor loss. An additional factor which also favors orange oil retention during granulation is that the emulsion being sprayed into the fluidized bed readily contacts dry powder particles in the fluidized bed. This results in a rapid reduction in moisture of the sprayed flavor emulsion, which would also decrease flavor losses.

Spray drying exhibited good retention of the orange oil, as was expected. Volatile retention is the result of the rapid formation of a dry skin on the surface of the atomized infeed emulsion, which inhibits the migration of the flavoring to the particle’s surface to be lost to the dryer air. One would expect the agglomerated SD powder to be lower in orange oil than the SD powder, since the SD powder was rewetted and then redried in the agglomeration process. The spray-dried sample underwent some flavor loss during this step. Thus, agglomerated samples were observed to have the lowest flavor load of all powders produced. This magnitude of orange oil loss during agglomeration was consistent with work reported by Buffo et al. [10].

### 3.4. Shelf Life Study

#### Effect of Processing and Storage on Limonene Oxidation

The effect of processing (spray drying, agglomeration and granulation) on the oxidative stability of the encapsulated orange oil is presented in Figure 6. The oxidation of encapsulated orange oil was generally monitored by measuring the formation of oxidation products: carvone and/or limonene oxides (we chose to monitor limonene oxides). It can be observed from Figure 6 that all samples had very low levels of oxidation products after manufacture. However, they all underwent oxidation on storage. The spray-dried samples had the highest amount of oxidation during the storage study. It is logical that the Agg powders were somewhat similar to the SD samples, since the SD powder served as the starting material for agglomeration. However, one may have expected the Agg powders to oxidize slightly more quickly than the spray-dried material due to the additional heat exposure during agglomeration. It is hypothesized that the agglomeration step (rewetting and drying) may have changed the oxygen permeability of the spray dry capsule surface, thereby offsetting any additional heat damage. The Gran samples displayed the lowest amount of oxidation, most likely partially due to the low heat exposure. This was consistent with the results of Burger et al. [6], who found Gran samples to be well protected from oxidation. The effect of heat exposure on the rate of oxidation of encapsulated orange oil oxidation has been documented by Anantharamkrishnan and Reineccius [12]. Nevertheless, it is interesting that the extra processing time used in the longer agglomeration times (25 to 90 min) had no significant effect on the level of oxidation, nor did the extra time used in granulation (30 to 90 min). We expect that was due to the low temperatures used in these two processes.

One would expect not only heat exposure during processing to influence oxidation rates, but also the physical properties of the particle, e.g., particle density, oxygen permeability and mean particle size. Spray-dried orange oil encapsulates were found to have a lower density than agglomerated or granulated materials. A lower density resulted in more particles present in a given weight of encapsulate, as compared to agglomerated and granulated products. More particles meant more surface area was available for oxygen permeation per unit mass/volume of encapsulate. Hence, more oxidation could take place inside the spray-dried sample, resulting in a higher limonene oxide content during the shelf life study.

A similar relationship exists for the mean particle size—a smaller mean particle size means, again, a higher surface area. Spray-dried products had the smallest mean particle size compared to agglomerated and granulated products; hence, it can be seen that the extent of oxidation was the highest for the spray-dried particles and decreased with agglomeration and, finally, granulation. The fact that granulation was performed under low heat treatment and produced large, dense particles, contributed towards lower oxidation during storage.

## 4. Conclusions

As noted in the Introduction, spray-dried flavorings have some potentially undesirable properties resulting from the small particle size and low particle density. These properties resulted in poor flow in manufacturing, slow dispersion in liquid solutions (when used as an ingredient in manufacturing or by the consumer) and dusty conditions in packaging and handling. The work presented herein evaluated the use of agglomeration and fluidized granulation to address these shortcomings. Results showed that granulation could product larger, more dense particles than the agglomeration of a spray-dried powder, which addresses the primary weakness of a spray-dried product. Additional benefits were that the granulated products showed better flavor retention and improved resistance to oxidation on storage than the combined spray drying/agglomeration product. The granulation process was also a one-step process rather than a two-step process of spray drying/agglomeration. The downside was that granulation is likely more expensive due to a limited production capacity, and flavor load in the finished product is limited by the need for an unflavored seed material.

## Figures and Tables

**Figure 1 molecules-27-01854-f001:**
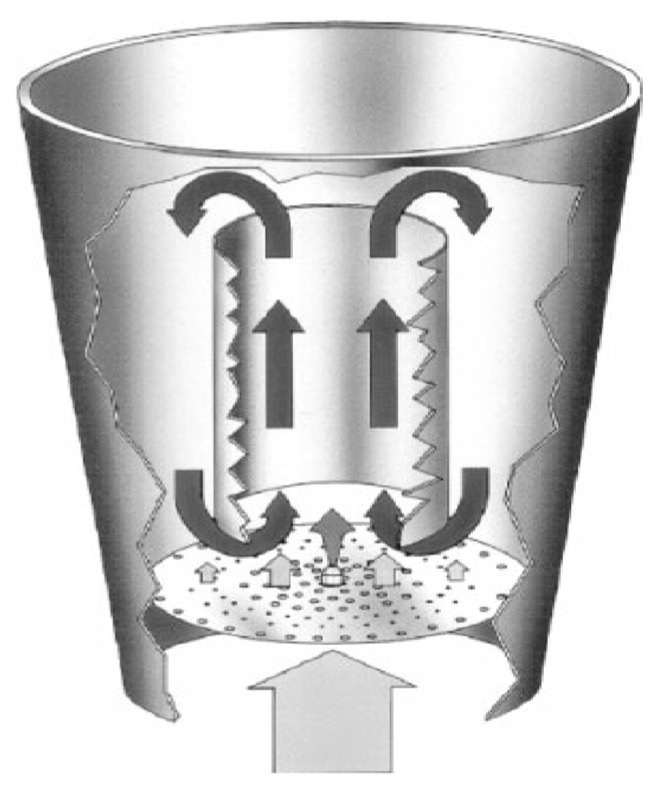
Diagram of bottom spray fluidized agglomerator (reprinted with permission from Glatt).

**Figure 2 molecules-27-01854-f002:**
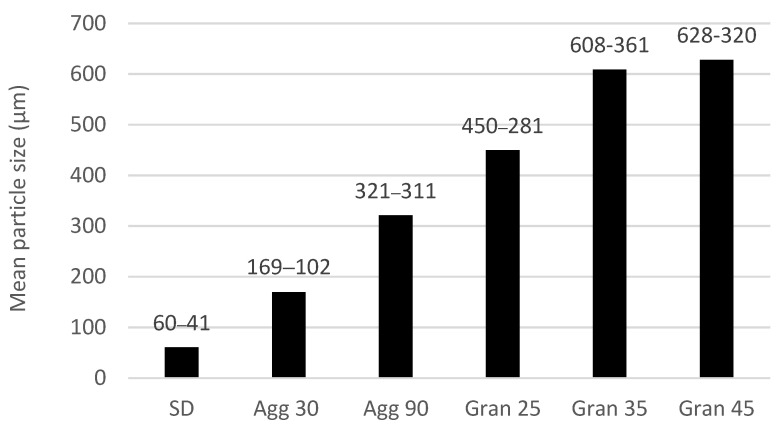
Influence of process (SD—spray dry; Agg xx—agglomeration min; Gran xx—granulation min) (values denote an average value of the mean particle sizes (µm) and standard deviations (µm) obtained for the duplicate runs).

**Figure 3 molecules-27-01854-f003:**
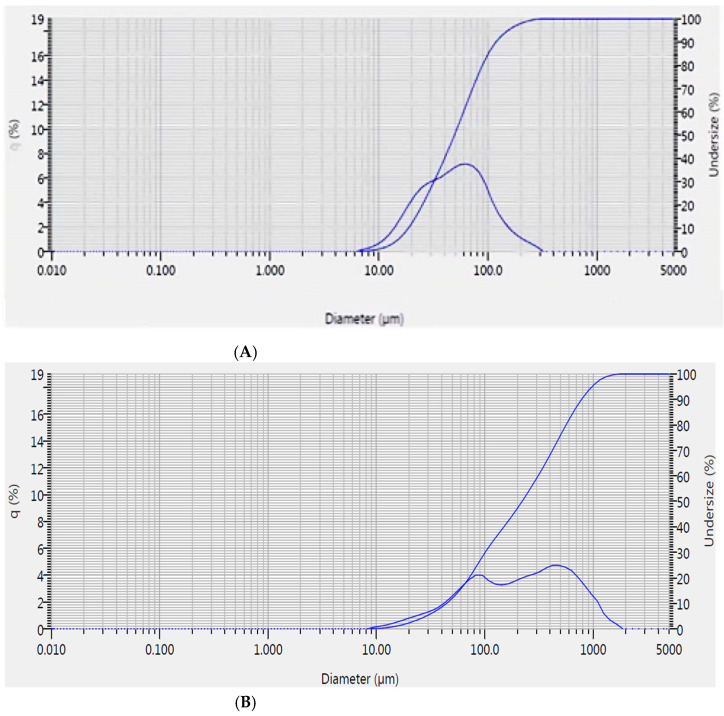
(**A**) Particle size distribution for spray-dried sample. (**B**) Particle size distribution for agglomeration 90 min run. (**C**) Particle size distribution for Gran 45 min run.

**Figure 4 molecules-27-01854-f004:**
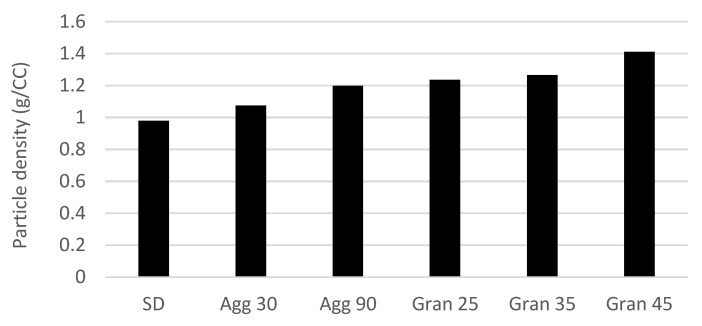
Influence of process (SD—spray dry; Agg xx—agglomeration min; Gran xx—granulation min) on powder density (values denote an average value of sample duplicates).

**Figure 5 molecules-27-01854-f005:**
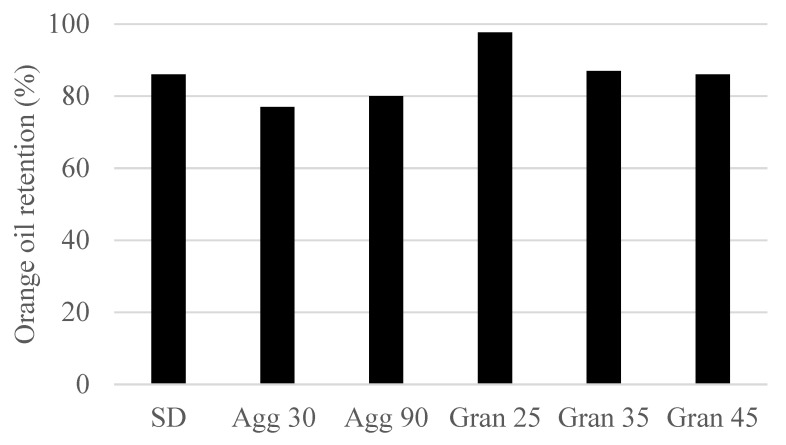
Influence of process (SD—spray dry; Agg xx—agglomeration min; Gran xx—granulation min) (values denote an average value of sample duplicates) on the retention of orange oil in the encapsulate.

**Figure 6 molecules-27-01854-f006:**
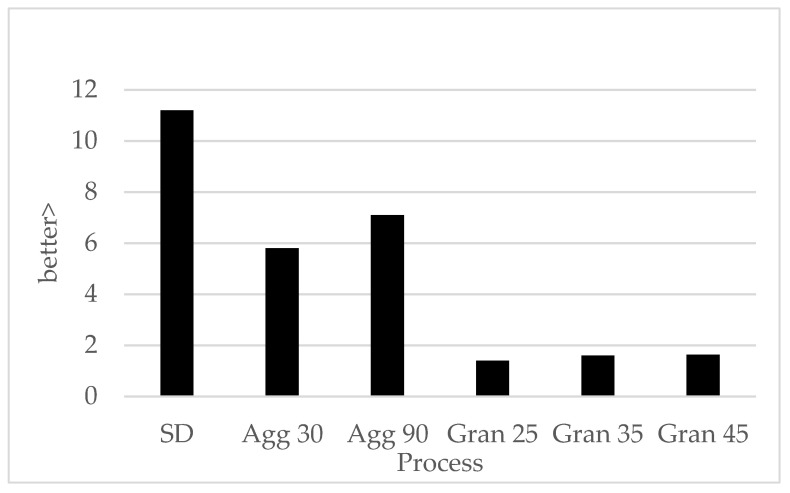
Extent of oxidation after 4 weeks storage at 45C. Oxidation is expressed as mg limonene oxide/g limonene (mgLO)/g limonene (L).

**Table 1 molecules-27-01854-t001:** Components of the orange oil emulsion used for encapsulation.

Components	Weight (g)
Capsul	360
Maltodextrin 15 DE	3240
Water	2945
Single-fold orange peel oil	740 (20% of carrier solids for SD and Agg samples) 925 (25% of carrier solids for granulation)

**Table 2 molecules-27-01854-t002:** Theoretical orange oil weight in 100 g powder for all samples assuming 100% retention.

Process	Flavor
Spray drying *	20.00
Agg run 30 min *	20.00
Agg run 90 min *	20.00
Gran 25 min **	12.18
Gran 35 min **	15.03
Gran 45 min **	16.30

* The infeed emulsion was 20% orange oil based on the carrier solids in the emulsion. ** The data on granulation considered 200 g unflavored seed plus the amount of fresh emulsion (25% orange oil based on solids) fed into the fluidized bed in the noted run time.

## Data Availability

Available from the research Director G.R.

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
