# Peer review of "Encapsulation of Orange Oil Using Fluidized Bed Granulation"

_molecules, 2022, doi:10.3390/molecules27061854_

Round 1

Reviewer 1 Report

The authors should mark the use of nanotechnologies as new direction and frontier in essential oil research  in the perspective of nutraceuticals, food and medical applications and previous examples should be given such as:

Vieira et al.. Sucupira Oil-Loaded Nanostructured Lipid Carriers (NLC): Lipid Screening, Factorial Design, Release Profile, and Cytotoxicity. Molecules. 2020, 25(3), pii: E685. doi: 10.3390/molecules25030685.

A graphical scheme of study approach should be given.

Introductory lines in section "Results and Discussion" should be inserted to better introduce and describe the different type of results.

The effect of processing on the retention of orange oil should be better described.

Results in figure 9 should be better explained.

Author Response

Thanks for your valuable comments, responses please see attached files.

Reviewer 2 Report

This study compared the physical and chemical properties of orange oil encapsulates produced by granulation and spray drying/agglomeration. Positive data were collected, but to meet the standard of publication, much is to be improved.

- All data in figures and tables are lacking standard deviation and statistical significance analysis.

- In order to have a systematic assessment of the physical and chemical properties of the encapsulates from SD, Agg, and FBG, more experiments such as water activity, and particle morphology, etc. should be conducted.

- In the storage study, the retention of flavor should also be tested.

Author Response

Thanks for your valuable comments, please kindly check the attached file.

Reviewer 3 Report

In this study, the granulation process involved in the production of larger and more dense flavoring particles was compared in terms of particle size, density, orange oil retention, and oxidative stability on storage. The manuscript presents a correct and well-designed experimental sequence to achieve the desired goal. In my opinion, the minor concerns for improvement of the paper given below could be considered by the authors before it could be acceptable for publication.

Minor concerns

The total weight of the emulsion can be corrected to 7,285g (line 121)

Line 128- please, correct 80oC

The morphology figure of the particles could be presented (line 203)

In my opinion, Fig 4, 5, and 6 could be displayed together as Fig. 4 -A, B, and C

Please, standard deviation bars can be provided to Fig 3 and 7

Author Response

Thanks for your valuable comments, please kindly check the attached files.

Round 2

Reviewer 2 Report

Not meet the standard of publication.
